# Harmonization of Sustainability Reporting Regulation: Analysis of a Contested Arena

Hammed Afolabi [ID], Ronita Ram [ID] and Gunnar Rimmel *[ID]

Henley Business School, University of Reading, Reading RG6 6AH, UK; hammed.afolabi@pgr.reading.ac.uk (H.A.); ronita.ram@henley.ac.uk (R.R.)
* Correspondence: g.rimmel@henley.ac.uk

**Abstract:** This paper presents the case for the sustainability reporting field as a contested arena and examines the behavior and the influence of the various actors, such as the Global Reporting Initiative (GRI), the Sustainability Accounting Standards Board (SASB), the International Integrated Reporting Council (IIRC), and the European Commission, including the European Financial Reporting Advisory Group (EFRAG) and the International Financial Reporting Standards (IFRS) Foundation in influencing the shape of the regulation in the arena. Drawing on the arena concept and documentary analysis, this study explores the dynamic in which each actor is attempting to change the rules within the arena and how this contributes to the harmonization and future direction of sustainability reporting. The findings of this study show that the actions and behavior of the various actors are premeditated and strategically calculated to maintain their influence, relevance, and defend their technical authority in the arena. The findings also suggest that sustainability reporting regulation is still far away from harmonization due to the perceived hegemony in the arena, and diversity in the overarching objective of the various actors and the inability of each actor to renounce its particular perspective and orientation. Insights are provided for policy makers on the urgent need to decide and reclassify the specific rules required in upholding the sustainability reporting arena.

**Keywords:** arena concept; EFRAG; GRI; harmonization; IFRS Foundation; IIRC; regulation; SASB; sustainability reporting

## 1. Introduction

Recently, there has been an exponential growth in the legislative efforts in pursuing a greener economy and society [1,2], which intensified the pressure for accountability of the social and environmental implications of companies' activities globally [3]. This has increased sustainability reporting prominence in the corporate setting [4] and particularly raises concern aboutthe regulatory framing that could ensure companies' adequate responsible behavior and quality of sustainability reporting beyond national borders [5]. However, over the last two decades, different non-state and private bodies (hereafter external actors) have emerged as transnational standard-setters and providers of guidelines for the benefitof shaping and improving the sustainability reporting regulatory arena. These bodies include the Global Reporting Initiative (GRI), the International Integrated Reporting Council (IIRC), the Climate Disclosure Standards Board (CDSB), the Task Force on Climate-related Financial Disclosure (TCFD) and the Sustainability Accounting Standard Board (SASB). Over time, they have issued myriad and diverse sustainability reporting-guidelines that are considered authoritative [6], and which continue to gain legitimacy despite these being voluntary and not legally binding.

Nonetheless, while these external actors continue to gain position in the sustainability reportingarena, different influential institutions have raised concerns aboutthe complexities and proliferation of sustainability reporting frameworks and standards, and call for harmonization of sustainability reporting practices [7–12]. These institutions include the

Accountancy Europe, the World Economic Forum, the International Organization of Securities Commission (IOSCO), the International Federation of Accountants (IFAC) and the Impact Management Project.Moreover, legislatures are also pushing for mandatory and harmonized sustainability reporting practices. For example, in the process of improving the Directive 2014/95/EU, the European Commission has given a mandate to the European Financial Reporting Advisory Group (EFRAG) to undertake preparatory work toward developing European sustainability reporting standards [13]. Consequently, the EFRAG has established a sustainability reporting standards board and is making significant changes to its governance and funding structure [14]. At the same time, in response to the various reports and harmonization calls made by different authoritative institutions [7,10,15,16], the International Financial Reporting Standards (IFRS) Foundation has also intervened and commenced work toward contributing to the sustainability reportingregulatory arena. Particularly, the Foundation's trustee has established the International Sustainability Standards Board (ISSB) to sit alongside the International Accounting Standards Board (IASB), to set globally accepted sustainability standards and to reduce the current complexities (and lack of comparability) present in the sustainability reporting arena [17].

As these developments unfold, the existing external actors have also taken considerable steps and actions in this context. For example, the SASB and the IIRC have merged and become the Value Reporting Foundation (VRF) (The activities of the IIRC and SASB are reviewed separately before the merger as the study aims to understand their behavior and influence in the sustainability reporting regulatory arena). Likewise, following the establishment of the ISSB, the IFRS Foundation has consolidated the CDSB into the ISSB and plans are ongoing for the potential consolidation of the VRF into the IFRS Foundation in June 2022 [18]. Hence, this demonstrates the degree towhich the sustainability reporting arena remains unstable and theuncertainties surrounding the harmonization of sustainability reporting regulation. Likewise, it suggests that there is a multitude of voices attempting to influence theshape of the sustainability reporting arena. Importantly, the entry of twoinstitutions with credentials in the accounting regulation setting: the IFRS Foundation and the EFRAG [19,20] (despite the existence of various external actors, such as the GRI and VRF) exemplify the contestation about the standards and frameworks that could become a norm in the arena, and how harmonization could be achieved. This, however, raises a crucial debate on how the existing external actors have influenced the shape of sustainability reporting regulation and how they may continue to exert their ruleswithin the arena, and what it could mean for the future direction of the sustainability reporting regulatory arena.

In agreement, researchers have drawn the connection between the acceptance of accounting standard setting and how technical characteristics of the standards and their providers are managed [21,22]. Therefore, this suggests that the actions andapproach of the external actors within the arena have the potential to influence the shape of sustainability reporting regulation, harmonization and its future direction.

Consequently, the main aim of this paper is to explore the behavior and influence exerted by various external actors within the arena, and how thesecontribute to the harmonization of sustainability reporting regulation. Furthermore, this study aims to examine the entry of the IFRS Foundation and EFRAG into thearena and to make sense of their impact onthe future direction of the sustainability reporting regulation. This represents a distinctive opportunity to show how the various external actors are influencing the shape of the regulation. Thus, to comprehend the behavior and influencing strategy of the external actors and what it could mean for the harmonization of sustainability reporting regulation, this study provides a documentary analysis of their development and activities in chronological order, particularly focusing on GRI, IIRC and SASB, (and the new influential institutions: the EFRAG and IFRS Foundation). These external actors are selected due to the evidence in diverse studies, surveys and reports acknowledging them as the issuers of the most rigorous and comprehensive guidelines for sustainability reporting internationally [23]. Following this, various documents pertainingto the activities and

output of the external actors, including the EFRAG and IFRS Foundation are examined. Likewise, the systematic literature review is conducted to analyze and complement the understanding of the institutional strategy of each external actor, and the development of their guidelines, frameworks, and standards and to help make sense of their behavior and implications for the future direction of the sustainability reporting arena.

Furthermore, Renn's [24] arena concept as developed by Georgakopoulous and Thomson [25] has been applied to explore how the various actors are behaving and influencing the sustainability reporting arena, and what this could mean for the harmonization and future direction of sustainability reporting.

Following the analysis and the application of the arena concept, it has been found that sustainability reporting takes place in a complicated environment, with complex interaction and strategy among the different actors, with the purpose of influencing and changing the rules in the arena. Additionally, the analysis points out that the actions and behavior of the various actors are premeditated and strategically calculated to maintain their influence and relevance, and defend their technical authority within the arena. Further, the findings present the case that harmonization of sustainability reporting regulation is not any nearer due to the extent of hegemony perceived in the behavior and influence of the various actors. Finally, the analysis suggests there is an urgent need for political institutions and rule enforcers to decide on or reclassify specific rules required in upholding the sustainability reporting arena.

Consequently, this paper contributes to the literature in several ways: first, this study extends prior studies on the diversity and harmonization of sustainability reporting, such as [6,26], by analyzing the historical behavior of the various actors and how this influenced the shape of sustainability reporting regulation. Second, this study sheds light on the focus of different actors in the arena [27,28], by documenting the diversity in their interests and the factors driving their differences. Third, theoretically, this study responds to the call for more research on how actors engage within an arena [25], offering new insights on different engagement tactics usedto pursue interests. Finally, the findings of this study offer insight for sustainability reporting policy makers on the potential implications of the various interactions within the arena on the harmonization and future direction of sustainability reporting.

The remainder of this paper is organized as follows. Section 2 presents the overview of the theoretical background that frames this study and the account of uncertainties and issues surrounding the sustainability reporting regulatory field as a contested arena. Section 3 discusses the method used in this study. Section 4 provides the documentary analysis of the key external actors and explores their engagement strategy and influence in the arena. Section 5 examines the potential implications of the IFRS Foundation and EFRAG on the harmonization and future direction of the sustainability reporting regulatory arena. Section 6 discusses the findings and considers the significance of this study for the future direction of sustainability reporting regulation and concludes the paper.

## 2. The Arena Concept

According to Renn [24], the arena is a "metaphor to describe the symbolic location of actions that influence collective decisions or policies." In other words, it represents the process of policy formulation and enforcement, and the pattern of interaction and strategy among various actors in a specific context. Thus, relevant in the arena are the actions and behavior of social groups and individuals that intend to influence policies or collective decisions [24]. Figure 1 presents the key elements of an arena.

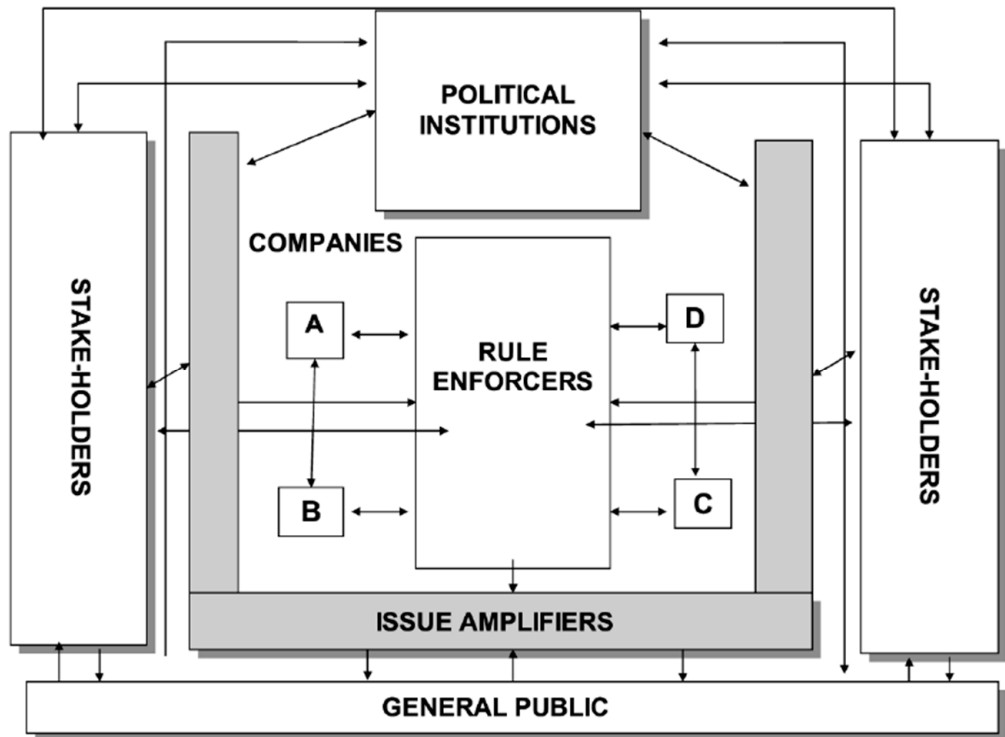

**Figure 1.** The Elements of an Arena. Source: Georgakopoulous and Thomson [25].

As shown in Figure 1, the centerof any arena consists of companies, and those tasked with enforcing the "rules" that they must fulfill. Additionally, it includes other actors, particularly the groups in society that intend to influence the guiding principles or policies within the arena. Renn [24] explains that often surrounding the central arena are wider stakeholders, (including the general public) who may adopt explicit support or anti positions (supportive or reforming) within the arena. Thus, stakeholders are refined to all the participants concerned with the specific issue in which an arena is based. In several arenas, rule enforcers are deemed to have "powers delegated to them by political institutions via legislation", and thus become the ultimate decision-makers [25]. As a result, all key actors and participantscommunicate their claims to this group and try to influence their decisions through various means. Thus, it is assumed that arena participants use social resources to pursue their objectives and maintain their influence within an arena. These resources include power, money, and professional and social influence [25].

Finally, issue amplifiers play a similar role to "theatre critics"as they observe actions on stage, communicate with the participants, interpret their findings and report to the audience [24]. Therefore, they mediate the relationship between the audience of wider stakeholders and the central arena participants. Importantly, they can influence arena dynamics through the mobilization of public support for a particular fraction ofthe arena. Thus, participants who play a significant role in drawing audiences and mediating the issues within the arena are identified as issue amplifiers, such as an independent supervisory body or campaigning NGOs [24]. Arena participants do not stay passive, but engage enthusiastically and have an underlying agenda to influence the outcome and decisions taken in a specific context. The arrows shown in Figure 1 signify the potential array of diverse engagements that can occur. Expanding this, Renn [24] submits that interactions in an arena may change the arena rules; small changes in the behavior or the strategy of the key actors and participants are capable of producing major changes in the outcome of the final decision-makers inthe arena.

*The Application of Arena Concept in the Sustainability Reporting Regulation Field*

By using the arena concept in analyzing the sustainability reporting regulatory sphere, different interactions can be taken into consideration and various participants can be differentiated. The arena metaphor is consistent with the typical characteristics surrounding sustainability reporting regulation, where the key sustainability issues and appropriate systemsto ensure consistent, comparable and quality sustainability disclosure globally remain debatable among different actors.

The term sustainability reporting has been conceptualized as the disclosure of non-financial aspects of a company's performance that are not captured within the mainstream of financial reporting to their stakeholders [29].Thus, it has become a conscious practice for companies to provide their stakeholders with material information about their actions on the environment, social and governance (ESG) issues, (such as climate change and human rights) that may affect their interests [30]. In fact, there is evidence of the rise in the momentum of companies towards sustainability disclosure, as the KPMG [23] survey results have shown that 96% of G250 companies now issue their sustainability report, compared to 45% in 2002 and 83% in 2008. Nevertheless, while sustainability reporting practice benefitsmay seem tremendous, it is argued that it is still far away from attaining equal eminence to financial reporting [7]. In agreement, researchers have shown that there is ambiguity in terms of the best regulatory approach to use; voluntary or mandatory, and the users' preferences in terms of corporate non-financial information disclosure [31,32]. Therefore, the approach of regulation to improve the quality of sustainability reporting practices beyond national borders remains contentious in literature [30].

However, there is evidence of the unrelenting efforts of legislatures to improvethe eminence of sustainability reporting incorporate settings. For example, in 1995, Denmark issued legislation for mandatory environmental reporting, covering about 3000 organizations, and was the first country to move towards a mandatory path [33]. Other countries, such as Norway, Spain, and the Netherlands have also taken considerable steps to improve ESG disclosure [34]. In the same vein, the UK and France, (particularly in the EU) have introduced a public mandate requiring all listed and large companies (with more than 500 employees) to disclose their ESG impacts [35].Thus, it can be argued that the provision of sustainability reporting is becoming mandatory, and in some countries, it is already legally mandatory. Additionally, during COP 26, government institutions and representatives across the globe echoed the need for global companies to address, and be accountable for the impact their activities have on people and the planet to tackle climate change (COP 26 is a major United Nations climate change summit that took place in Glasgow in November 2021 to discuss how climate change will be tackled and to agree on global and national targets (UN, 2022)) [36].

While the above may signal a move towards steady sustainability reporting regulation, there is no internationally agreed framework or consensus on the common pattern in which a company's impact on ESG should be disclosed globally. This, however, reinforces the credibility of various external actors, (such as the GRI and SASB) as the mediator mobilizing public support and their awareness, and increasing the legitimacy of sustainability reporting by providing continual guidance for companies to follow. It can be argued that the responsibilities of these external actors, under the boundary of their engagements, include offering beneficial components and directions for businesses to follow. However, they have also been called "self-serving, and self-absorbing and very rarely systems-changing" [37].Further, prior studies have shown that the significant hindrance to high consistency, quality and comparability of sustainability reporting include differing ESG priorities in various countries [38,39], and where mandatory reporting is not being enforced [40,41]. This also represents the premise of other actors (pressure groups) pushing and campaigning for harmonized sustainability reporting [7,8,11]. The contestation that remains is the availability of diverse information from various quarters, proposing different paths for the future direction of sustainability reporting regulation. Additionally, whether the harmonization is plausible and which version of initiatives could become the norm to ensure comparability and consistency of sustainability reporting globally. Therefore,

with the number of key external actors involved in the sustainability reporting regulatory arena, and the intervention of the IFRS Foundation and EFRAG the controversial question that remains is: are the various institutions working wholly towards harmonizing and preserving the core premise and tenets of sustainability reporting or are they competing to be the most influential shaper of sustainability reporting regulation?

Consequently, the current situation of sustainability reporting regulation reveals the need for an engagement with the arena framework. First, Renn [24] emphasizes that information flow and the strategy of the engagements of various actors are significant in an arena. This corresponds to the reality of the sustainability reporting regulatory arena, since decisions or outcomes of the arena may be slightly influenced by the actions and prepositions of various actors. Therefore, the arena concept offers a skeletal frame that allows the representation of different engagements and interactions within an arena. Second, through the application of the arena concept, Georgakopoulous and Thomson [25] investigate the interactions among the main participants in Scottish salmon farming and their use of social reporting. The findings suggest that different actors may choose diverse strategies to interact with each other, and the interaction may change the arena rules. The interaction of these strategies may even have an undesired outcome that does not comply with any of the actors' objectives. Therefore, Georgakopoulous and Thomson [25] recommend the need for further studies that "prepare shadow or silent account of the arena discourse, rather than single fragmented entities" to examine the depth of participants' engagement tactics and identify their various contradictions and competing motivations.

This study, therefore, explores the sustainability reporting environment and identifies the main actors within the specific case of the sustainability reporting regulatory arena (companies, rule enforcers, political institutions, issue amplifiers, categories of stakeholders and the general public). Consequently, the amount of influence of each actor, their objectives and interdependencies are explored and analyzed. Hence, drawing from Figure 1, the following actors can be identified and analyzed below:

**Companies**: In the context of this paper, "companies" as actors in the arena, are all businesses that are expected to disclose their impacts on ESG, regardless of their size and location.

**Rule enforcers**: Due to the proliferation of frameworks in the sustainability reporting arena, the rule enforcer includes regulators who have the authority to enforce specific regulations for companies to follow, such as the European Commission. Their influence is significant since they are the ultimate decision-makers and their roles include ensuring adequate frameworks are complied with for quality sustainability reporting. Further, because the EFRAG is working under the authority of the European Commission, it will be regarded as a rule enforcer.

**Political institutions**: These include government institutions, such as the European Union member state and other country-specific political institutions. Their influence is also significant as they have been identified in prior studies as having considerable influence in the regulatory framing of sustainability reporting, especially in Europe [42].

**Issue amplifiers**: In the case of this study, the issue amplifiers are represented by external actors, such as the GRI and SASB. They are the ones playing the role of mediator in the arena by mobilizing and providing various guidelines that are expected to improve sustainability disclosure within companies. Their influence is significant as their perception ofsustainability reporting regulation could impact the extent of a company's sustainability disclosure and lead to a change in the rules within the arena.

## 3. Method

To explore the behavior of the selected actors, and to help make sense of their contribution toward the harmonization and future direction of sustainability reporting regulation, this study is based on documentary analysis and a systematic literature review.First, for the documentary analysis, public releases of the various actors pertinent to this study have been used to examine the extent towhich the sustainability reporting regulatory sphere

has become a contested arena, including the behavior of these actors and how they are influencing the shape of sustainability reporting regulation.These public releases are related to sustainability reporting guidelines, frameworks and standards issued by the external actors (GRI, IIRC, and SASB), and publicly available documents that illustrate the works of the EFRAG and the IFRS Foundation in the arena (see Table 1). This study relies on these documents because they reflect the conventional ways the various actors are thinking, and reinforce the pattern in which they are contributing to the regulation in the arena.

**Table 1.** Documents from the key actors examined in this study.

| Key Stakeholders | Documents | Year Issued |
|---|---|---|
| **GRI** | • G1 Guidelines<br>• G2 Guidelines<br>• G3 Guidelines<br>• G4 Guidelines<br>• The GRI Standards: The global standards for sustainability reporting<br>• Revised Standards<br>• GRI Universal Standards Project-GSSB basis for conclusions | • 2000<br>• 2002<br>• 2006<br>• 2014<br>• 2016<br>• 2016<br>• 2021<br>• 2021 |
| **IIRC** | • \<IR\> Framework<br>• \<IR\> revised Framework | • 2013<br>• 2020 |
| **SASB** | • SASB Conceptual Framework<br>• SASB Exposure draft (revised framework) | • 2017<br>• 2020 |
| **European Commission/EFRAG** | • EU public consultation regarding the proposal by the European Commission for a regulation<br>• Proposals for a relevant and dynamic EU sustainability reporting standards (PTF-NFRS)<br>• Potential need for changes to the governance and funding of EFRAG<br>• Legislative proposal for a corporate sustainability reporting directive (CSRD) | • 2020<br>• 2020<br>• 2021<br>• 2021<br>• 2021 |
| **IFRS Foundation** | • IFRS Foundation consultation paper on sustainability reporting<br>• Responses to the consultation paper on sustainability reporting<br>• IFRS Advisory Council agenda and agenda papers<br>• Trustees announce steps in response to broad demand for global sustainability standards<br>• Strategic direction and further steps based on feedback to sustainability reporting consultation | • 2020<br>• 2020<br>• 2021<br>• 2021<br>• 2021 |

The period of analysis was from 1997 to 2021, which captured major revisions and developments in each of the frameworks and standards released by the GRI, IIRC, and SASB, including the recent developments made by EFRAG and IFRS Foundation. This time frame was chosen because the first sustainability reporting regulatory institution, the GRI wasestablished in 1997, and the analysis includes other relevant documents and studies released in2021.The documents were downloaded from the websites of the selected actors accordingly, and imported into an excel sheet and categorized according to the issuer. Consequently, the guidelines and standards issued by the various external actors were studied carefully to comprehend their development, shape and trajectory over the years. The documents were meticulously screened based on five major categories, which include (1) year issued (2) mission (3) primary objective (4) sustainability context, and (5) target audience. This enabled the authors to comprehend the pattern, in which the various

guidelines and standards are developed and shaped, and the approach of each actor in regulating the sustainability reporting arena.

Second, the systematic literature review (SLR) is used specifically tosupport the under-standingof the institutional strategy and behavior of the external actors (GRI, SASB and IIRC) and the implication of their respective guidelines, frameworks, and standards. The SLR is ap-propriate as it possesses the potential to provide verifiable answers to narrowly structured questions [43]. Therefore, the SLR is used to reinforce and complement the understanding of the behavior and tactics of the selected external actors and explore how they may be affecting the shape of regulation within the arena. The review passesthrough four pri-mary processes; (1) search (2) collection (3) screening and (4) analysis. Figure 2 offers a visualization of the SLR process. The search process represents the first phase, and thus the sources were identified first. Five major databases were used, which include Scopus, Google Scholar, Wiley Online Library, Emerald and Web of Science for investigating scien-tific published papers. The keywords used for the search (as described in the SLR) were "sustainability reporting", "GRI" "IIRC","SASB", "institutional strategy", "framework", "guidelines", "standards", "development", "practical implication".They were entered into the selected databases accordingly and the publication time frame was limited to the same period as the documents gathered (1997–2021), which resulted in 2450 articles.

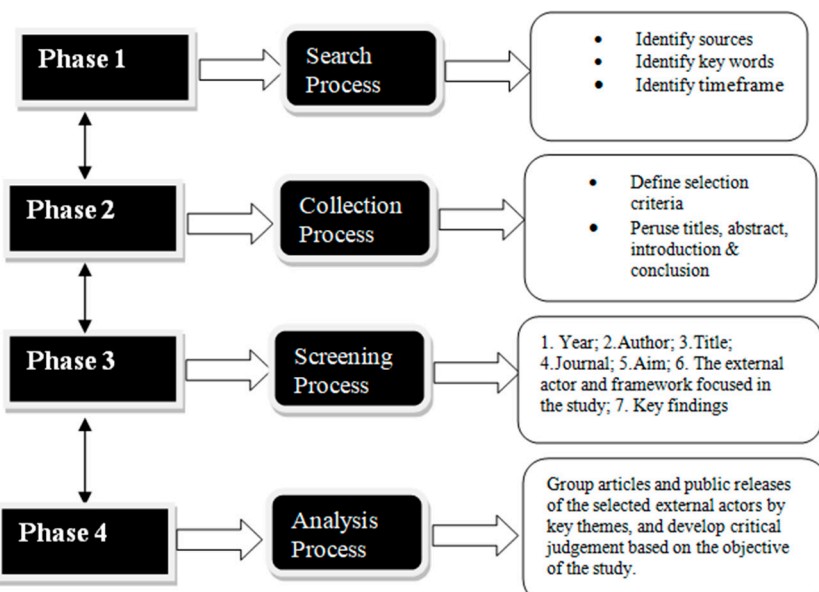

**Figure 2.** The Systematic Literature Review-Research Process.

The second phase of the review is the collection process. In selecting the articles relevant to this study, the search for the articles covered two thematic areas, which include the articles that explored:

- the background of any of the selected external actors (GRI, SASB and IIRC) and concerned with the practice and implication of their guidelines and frameworks and standards issued.
- the approach used by any of the selected external actors' in mobilizing provisions and regulating the sustainability reporting arena.

These themes were selected because the overarching objective of this study is to un-derstand the behavior and influence of the external actors (GRI, SASB and IIRC), and to help make sense of how they are affecting the shaping of the sustainability reporting regu-latory arena. Thus, the themes helped to identify the articles that explored the institutional strategy and implication of the various guidelines, frameworks and standards issued by any of the selected external actors. In this sense, Denyer and Tranfield's [44] explanation of SLR as "a specific methodology that locates existing studies, selects and evaluates con-

tributions, analyses and synthesizes data, and reports the evidence in such a way that allows reasonably clear conclusion to be reached about what is and is not known" was followed. Therefore, an initial review of each abstract, title, introduction and conclusion was conducted by one author, and articles that were not concerned with any of the thematic areas were eliminated. The complete list of the inclusion and exclusion criteria is provided in Table 2.

**Table 2.** Inclusion and Exclusion Criteria for Literature Search.

| | |
|---|---|
| **Inclusion Criteria** | • Scholarly published/Peer reviewed articles<br>• Written in English Language<br>• Published anytime from 1997–2021<br>• Addressed at least one of the two thematic areas |
| **Exclusion Criteria** | • Do not address any of the thematic areas of investigation<br>• Only mentioned regulation in citation<br>• Book chapters;<br>• Written in a language other than English;<br>• Conference papers and bachelor's, master's and PhD theses |

Following the application of the criteria as shown in Table 2, the search resulted in 45 articles, published in more than eightjournals (for example, Accounting Auditing and Accountability Journal, Critical Perspectives on Accounting, Social and Environmental Accountability Journal, British Accounting Review, Journal of Business Ethics and Meditari Accountancy Research). Therefore, articles that revealed the background and evolution and institutional strategy, including the implication of any of the frameworks, guidelines and standards issued by the GRI, SASB and IIRC are selected. The third phase of the review is the screening process. In line with prior studies [45,46], seven categories were created to screen the articles' content: (1) the publication year, (2) author(s), (3) title, (4) journal, (5) the study aim, (6) external actor and framework studied (7) key findings. This information was sorted out using an excel spreadsheet, which was subsequently used in the analysis process. See Appendix A Table A1 for the summary of the key findings of the articles selected for a sample.

The final phase of the review includes analyzing the articles together with the documents collected.This was conducted to corroborate the understanding of this study on the strategy, influence and behavior of the external actors examined in their various public releases. Therefore, the articles were grouped under four categories [47]; (1) distribution of articles by year, (2) main focus of the articles in terms of objectives/questions relevant to the external actors (3) analysis time frame and (4) findings. This process helps to comprehend and aligns the findings of the articles with the various documents examined, and thus reinforces the understanding of the behavior and influence of the external actors within the arena. Consequently, the documents analyzed are presented and discussed chronologically in Section 4 to explore the behavior, influence and engagement tactics of the external actors.Additionally, the documents are used in Section 5 to explore the role and implication of the entry of the IFRS Foundation, European Commission and EFRAG into the sustainability reporting arena, and how this contributes to the harmonization and future direction of sustainability reporting regulation. Further, the articles selected are cited and their analysis wasused in Section 4 to support the understanding of the strategy and influence of the external actors (GRI, SASB and IIRC), which help to make sense of their behavior and how it is affecting the shape of the sustainability reporting regulatory arena.

## 4. The Issue Amplifiers and Wider Stakeholder Engagement-Analysis of Their Influence in the Contested Arena

To explore and comprehend the way the key external actors are behaving and engaging within the arena and how they may be influencing the shape of sustainability reporting regulation, this study provides a documentary and comparative analysis of their emergence, guidelines, and standards, due process, and outcome. Therefore, it becomes imperative to analyze the pattern of their developments to examinehow they are mobilizing provisions for sustainability reporting regulations and their engagement tactics. Table 3 gives a snapshot of the timeline pertinent to their evolution and development over the years.

**Table 3.** The Evolution and Development of the Issue Amplifiers.

| Date | Key Activities and Developments |
| --- | --- |
| 1997 | GRI was established |
| 2001 | GRI (G1) was launched |
| 2002 | GRI (G2) was launched |
| 2006 | GRI (G3) was launched |
| 2010 | IIRC was founded |
| 2011 | SASB was founded |
| 2013 | International Integrated Reporting <IR> Framework |
| 2014 | G4 was launched |
| 2016 | GRI standards launched |
| 2020 | SASB published exposure draft for the revision of its conceptual framework |
| 2021 | IIRC and SASB merged and become Value Reporting Foundation |
| 2021 | <IR> Framework was revised |
| 2021 | GRI standards were revised |

### 4.1. The Relevance and Influence in the Sustainability Reporting Arena—How Do the Issue Amplifiers Play?

The GRI was established in 1997, its roots emanated from the Coalition for Environmentally Responsible Economies (CERES) and the Tellus Institute (with the involvement of the United Nations Environment Programme) [48]. The GRI development began when its founders (Allen White and Bob Massie from the NGOs sector) discovered a core tension within the space of social reporting, particularly relating to competing institutional logic between "civil regulation and corporate social performance" [49,50]. These controversies remain: the "logic of civil regulation" considers social reporting a means of empowering civil society groups to be involved actively in corporate governance, while the "logic of corporate social performance" identifies the instrumental value of social reporting to the investor community, corporate management, including consulting and auditing firms [50]. Despite the lackof formal authority and substantial resources, White and Massie managed to create an alliance between NGOs and businesses by campaigning for "win-win judgement" which considers the prior two logics as complementary rather than incompatible [50].

Consequently, the win-win judgment preposition garnered immense debate, especially in the corporate sustainability literature of the early 2000s [51]. However, the overarching premise of most of the studies remains that the GRI founders' judgment will improve the possibility of addressing social and environmental concerns in ways that can advance corporate profitability [52]. In fact, their judgment is considered as a mechanism that can generate confluence among different stakeholder groups, rather than advancing conflict of interest [53]. Therefore, the GRI's core strategy and aim are assumed to be closely related to the concept of ESG and triple bottom line reporting. In agreement, Levy et al. [50] interviewed the GRI's co-founders about their institutional strategy and submitted that the GRI's co-founders' belief is that standardized information on ESG is significant for ranking and benchmarking and, in turn, empowers civil society organizations to demand greater corporate accountability from business.

The GRI remains the pioneering sustainability reporting institution and has developed various versions of its guidelines, with the first guidelines (named G1) introduced in 2000, the second (G2) released in 2002, and the third generation of the framework (G3) launched in 2006 [23,54]. Its various guidelines are identified as de-facto guidelines due to its approach of institutionalizing multi-stakeholders' interest in the development of the guidelines [55]. The key industries representing the core constituencies of the GRI's network include investment institutions, civil society organizations, labor and mediating institutions and business [56]. Following the review of the various documents pertaining to the GRI, this study identifies that the diverse GRI guidelines are mostly tailored towards a company's sustainability reporting, and accountability, especially the G3 introduced "sustainability context principle", which required companies to communicate the extent of their impact on ESG. Further, G3 requires companies to self-declare their application level (either A+/A/B+/B/C+/C) of the framework or engage a third party to verify their self-declaration.

However, while the motivation of the GRI seems ethical and can be perceived as a strategic initiative to foster corporate citizenry, its ability to foster greater accountability remains contentious. Expanding on this, it can be argued that G3 has a credibility gap due to its extensive scope limitation for verification of a company's compliance. In addition, Isaksson and Steimle [57] analyzed five reports in the cement industry prepared using the GRI framework. They found that the guidelines are inadequate to promote disclosure pertaining to an organization's capacity to be sustainable. Similarly, Boiral [58] examined the non-financial reports of 23 energy and mining companies issued in 2009, from 14 countries that had received "A or A+" application rating from the GRI, and found that 90% of significant negative events were not reported. This creates an avenue for guidelines that can promote not only a company's measurement of the historical impact of ESG but also sets how it affects the key components and different interrelated dimensions of the company's activities [59].

### 4.1.1. The Emergence of the IIRC and SASB—How Do They Behave in the Arena?

Following the proponents harnessed in the previous section, this study identifies that there is a gap in the GRI, which arguably, can be connected to the motivation for the establishment of the IIRC in 2010 by the GRI and the Accounting for Sustainability Project. Following the joint press release by the two institutions, it was emphasized that the "the intention is to help with the development of more comprehensive and comprehensible information about an organization's total performance, prospective as well as retrospective, to meet the needs of the emerging, more sustainable, global economic model" [60]. Therefore, it can be argued that the initial underlying agenda of establishing the IIRC is to help save the planet. The vision of the council is relatively connected "to communicate company financial and non-financial information to a broad range of stakeholders (integration of financial reports and sustainability reports) through the addition of soft indicators of companies' performance, otherwise called 'integrated thinking' " [60]. At the time of writing, the IIRC comprises sixty-six members from diverse industries, which include the CEOs of the 'Big Four', the heads of the IFAC, IOSCO, IASB, GRI, representatives from different British Professional accountancy bodies, and other multi-national institutions (such as the World Economic Forum and The World Bank). In 2011, the IIRC introduced the <IR> consultation draft that combines ESG and financial reporting into a single report.

Nevertheless, much later, the IIRC started to gain recognition in the sustainability reporting arena and, the SASB emerged as another independent standard-setter in July 2011. Established by Jean Rogers, the SASB aimed to help companies manage, identify, and communicate financially-material sustainability information to various investors in a useful and cost-effective format for decision-making in the Securities and Exchange Commission (SEC) filings in the US [61]. At the time of writing, the SASB is governed by its Foundation Board of Directors which consists of eighteen members from various industries, particularly banking and accounting firms. Subsequently, the SAB develops sustainability accounting

standards for 79 industries, with 11sectors through its Standard Board. Guidance is offered on 434 provisional disclosure topics to help companies when assessing their exposure to financial risks. Thus, the ESG issues focused on are around five dimensions: social capital, human capital, environment, business model and innovation, leadership and governance [62]. While it may appear like the SASB is pursuing a similar path tothe IIRC, the SASB was diligent enough to clarify its mission and intent to be a standard setter and not a guidelines provider [62]. Particularly, it elucidates its adherence to the concept of financial materiality as recognized by the various financial accounting standard setters, such as the Financial Accounting Standards Board (FASB). In this sense, the financial materiality of the non-financial issues appears to be a core element of the SASB.

Subsequently, in December 2013, (and after two years of consultations and feedback from various respondents), the IIRC published the international guidelines called the integrated reporting framework (<IRF>) [63]. Following the review, the<IRF> remains a principles-based document that supports integrated reporting, which focuses on "concise communication addressed to the internal and external stakeholders that illustrates how an organization's strategy, governance and perspectives make it possible to create value in a short, medium and long-term" [63]. Particularly, it fosters organizations drawing on diverse capitals as inputs and explains, through their business activities how they are converted to output to benefit their stakeholders, including how the capitals are managed to create wealth. These capitals include financial, manufactured, intellectual, human, natural, social and relationships. Following this, the IIRC clarifies that the aim of an integrated report is "to explain to providers of financial capital how an organization creates value over time" [63]. Therefore, this signals a difference in the initial agenda of establishing the IIRC and its proposed plan for sustainability reporting.

There are diverse criticisms leveled at the <IRF> in the literature, which includes its desertion of other key stakeholders for the financial capital providers only [27], other potential issues pertinent to the assurance aspects of integrated reporting, and the high possibility of having under-substantial narratives due to the subjective concept of the six capitals [64]. Therefore, it can be argued that <IRF> offers to be an extension of financial reporting, which looks like a good way forIIRC to signal its focus and brand itself. In this way, the GRI appears to be losing its influence and market share in the sustainability reporting space, which may be connected to the development of the SASB sector specific standards, which target different audiences and industries. In agreement, there is empirical evidence suggesting that <IRF> is engaging stakeholders more, particularly from the financial market (such as investment funds, banks, investors) and, therefore, has the potential of improving investment decision making [65,66], reinforcing its focus on "shareholder value" [67].

4.1.2. The GRI's Further Development and Activities of Other Actors

Following the various contributions of the SASB and IIRC in thearena, particularly with their approach tocreating a new path for sustainability reporting practice, the GRI appears to be losing its market share and relevance. Subsequently, in May 2014, the GRI published its fourth generation guidelines (G4) after several consultations with various stakeholders. Debatably, this appears as a way of rekindling its relevance and matching up with the SASB, particularly based on 'materiality'. Materiality is related to when an "organization prioritizes reporting on those topics that reflect its most significant impact on the economy, environment, and people, including impacts on human rights" [68]. In this sense, similar to the SASB standards, the G4 gives significant emphasis to "material aspects", with the aim to aid organizations in a new standardized approach to sustainability reporting. Additionally, to provide pertinent insights to present sustainability information in various formats of reporting, such as an integrated report, annual report and sustainability report [69]. The G4 offers two documents for reporting companies: the principles manual and the implementation manual for the guidance and preparation of non-financial reports by companies of different sizes and sectors. Importantly, it has "46 Aspects", which details the number of ESG material impact reporting companies can consider [69]. Further,

it has a requirement of 'Aspect Boundaries', which the GRI interprets as a requirement to explain where an impact occurs and describe who it affects, either within their entity or outside, including the geographical location.

However, there is no clarity on how to determine materiality, therefore, companies can make materiality whatever they want it to be. This means performance indicators (Aspect) focus on reporting actions and not impacts, especially indirect impacts, and there is a lack of detail to explain how stakeholders'inputis converted. Complementing this, Kumar et al. [70] analyzed the sustainability reports of the top 10 banks in India on the G4's parameters and found that metrics (such as equal remuneration) are omitted, and the engagement focus of the stakeholders of most of the banks is relatively weak. This evidence is also observed in the hydroelectricity provider (industry) in Quebec, Canada [71]. This demonstrates the degree towhich the G4 lacks the detail ofa company'smeasurement of itsindirect impact. Hence, it can be argued, that the G4 encourages companies to be more drawn to explain what they did, rather than where they made a difference to the environment and people's lives. Arguably, these limitations explain the establishment of the Global Sustainability Standard Board (GSSB) by the GRI. In 2015, the GSSB was created to direct the preparation and development of the GRI standards [72]. At the time of writing, the GSSB consists of 15 members from different regions and sectors, with diverse technical expertiseand experience from multi-stakeholder perspectives from various constituencies(especially the UK, Australia, USA, Belgium, HongKong and South Africa),and in the independence of the GRI board, the funding sources and the stakeholder council [72,73].

Consequently, in May 2016, the GSSB introduced the first globaland voluntarily accepted sustainability reporting standards, which appears an improvement and transition from the G4 guidelines. The standards were launched with an underlying motive if applied consistently across the globe, they can provide all stakeholders and companies the capacity to compare the impacts of diverse sustainability reporting systems, (because of its compatibility with a wide range of existing frameworks, particularly the SASB) [74].

Following the review of the GRI standards, whilethis study identifies that they possess many similarities with the G4 (particularly in terms of materiality, greater clarity, structure and personalized language), we foundsome other significant changes and developments. First, there are now requirements and specific guidance for companies to disclose how materiality is determined [74]. Expanding this, the GRI replaces the term 'Aspect' introduced in the G4 with 'Topics' in the standards, to specify various material issues, thus, companies are now expected to disclose their impact. The GRI has carved down the topics to 35 from an initial 46 aspects in the G4 by reshuffling different specific ESG issues to give better clarity. For instance, the three supplier assessments for society, labor and human rights have been pushed together into a single supplier social assessment topic. Additionally, anti-corruption (now, GRI 205) that is under social topics in G4, is now categorized under economic topics.

Second, contrary to the G4's interpretation of the "Aspect Boundaries, which were previously interpretedas where the impacts occur", the GRI has, however, clarifiedthe "Boundary" to relate to companies that cause the impact, regardless of where the impact occurs [74]. In this sense, companies are now required to reveal the impact cause of any of their entities (both inside and outside), which means they are now responsible for their suppliers' impacts. For example, the GRI 409: forced/compulsory labor, is a material issue, hence companies must assume responsibilityforits impact even if its impact occurs at the supplier's factory because the supplier is hired by the company. Further, the standards implemented specific standard-setting practice lingo to indirectly ensure complete compliance and credibility. For instance, there are disclosures that read "you shall report-means required to be in accordance", and "you should/can report-means guidance or recommended". Hence, this eliminates the likelihood of companies misinterpreting the materiality matrix and will help to promote consistency of sustainability reporting materiality, which is lacking in the SASB standards, particularly the <IRF>. Thus, the GRI continues to gain momentum and influence the authority of sustainability reporting

globally. For example, the KPMG [23] survey results have shown that 73% of the largest 250 global companies and 67% of the largest 100 firms from 52 countries are using the GRI standards.

Although the standards fail to address how the impact of reporting organizations across the entire value chain can be measured, including the impact of human rights-related activities and business relationships. The implications of this have also been demonstrated in prior studies [75,76]. Nonetheless, the GRI does address the limitation of its guidelines (G4) and has a consistent approach to engaging with and preserving the core tenets of sustainability reporting.

In 2017, the SASB published its conceptual framework (The SASB conceptual framework is currently being revised and this study also reviewed its exposure draft (published in 2020)), which defined its objectives, basic principles and concepts of its standards. Similar to the GRI, the SASB emphasizes the transparency, rigor, and inclusivity of its standard setting process based on materiality, and evidence, and is financial market informed [61]. However, while comparing this with the GRI standards, this study identifies that SASB draws its material topics around issues that have direct financial impacts and risk (issues that can affect a company's ability to create long-term value). Importantly, its emphasis is more in line with the US context Thus, the SASB's overriding goal is for its disclosure to be shown in the company's filed documents, such as the annual report on Form 20-K and 10-K for US public companies [61]. This is, perhaps, a branding strategy to distinguish itself from the GRI and the IIRC.

However, in 2020, the SASB published its exposure draft for the revision of its conceptual framework. In this draft, the SASB appears to move beyond its previous alignment with US law and now positions itself as a global standard setter. Particularly, the SASB explicitly declares its harmony with other global standard setters that prioritize capital providers and investors' interests, such as the IIRC and IASB in the conceptualization of its materiality. Thus, SASB now conceptualizes materiality as information that is financially material and:

"if omitting, misstating, or obscuring it could reasonably be expected to influence investment or lending decisions that users make on the basis for their assessments of short-, medium-, and long-term financial performance and enterprise value" [77].

This illustrates that only the issues that impact or affect a company's ability to create value will be considered, and therefore echoes the concerns about the relevance of other sustainability issues. Complementing this, in 2021, the SASB and IIRC merged and became the Value Reporting Foundation (VRF), a unified organization intended to provide investors and corporations with a corporate reporting framework across the full range of enterprise value drivers and standards [78]. Subsequently, after consultations with the public, the IIRC released its revised <IRF> framework. Following this review, it has been identified that the framework addresses the ambiguity surrounding its materiality, any issue that could have "extreme consequences" would be material even if it is highly probable not to occur [79].

At the same time, after several public consultations, the GRI revised its standards (The GRI released its revised universal standards in October 2021 (effective from January 2023)), which elucidates different principles for classifying and defining report quality and content, including guidance on materiality. As this study previously noted the weakness of the standards, the GRI has now made a significant shift from defining topics based on "importance to stakeholders" to the level of significant impact on the "planet (environment), people and economy", including the impact on human rights [80]. The significant material topics companies should now consider are the most important societal impacts across their value chain.

Following the proponents above, it can be argued that the issue amplifiers are making considerate efforts to mediate among other actors in the arena, which include the political institutions, companies, rule enforcers and other stakeholders. However, drawing from the knowledge of the arena concept (as discussed in Section 2) and the pattern in which they

are addressing and mobilizing the provisions of sustainability reporting, there is obvious evidence of their behavior and influence on shaping sustainability reporting regulation. Importantly, with the various gaps they addressed in the arena at each specific time, it can be argued that they possess a distinctive perspective toward sustainability reporting, which formulate their technical authority and interest within the arena. More specifically, this review demonstrates that the intervention and the strategy of the issue amplifiers are premeditated, as the technical authority they occupied individually emanates from the perceived limitations and weaknesses of others. Thus, the next section takes on the task of contextualizing their key controversies and interest in the arena.

*4.2. Contextualizing the Diversity and Interest of the Issue Amplifiers-What Are the Key Issues?*

Following the history and the different issue amplifiers' reviewed documents in the previous section, Table 4 elucidates their core interest and controversies (The GRI analysis is based on the new revised standards).

**Table 4.** The Issue Amplifiers' Core Interests and Diversity.

| Diversity Metrics | GRI (Standards) | IIRC | SASB |
|---|---|---|---|
| Scope/Principles | Standards-Fosters comparability and consistency | Principles-based system-Fosters fundamental assumption | Rule-based system-Evidence based |
| Primary Audience | Multi-stakeholder | Financial capital provider | Investment community |
| Materiality | Double materiality-Emphasizes disclosure of all aspects that reflects organizational ESG impacts, and that can influence the multi-stakeholder's ability to assess organizational performance | Financial materiality-Disclosure of information on all aspects that affect the organization's capacity to create value over the short, medium, and long-term substantively. | Financial materiality-communicate financially-material sustainability information to various investors |
| Core Implication | To understand a company's impact on people, plant and other stakeholders from organizational activities | Fosters an organization's integrated thinking in creating value | World's impact on a company's financial portfolio |
| Key Stakeholders Involved (Pressure Groups) | • Civil-society Organizations<br>• Business<br>• Labour and mediating institutions<br>• Academics<br>• Investment Institution | • World Bank<br>• Accounting professionals<br>• Investment institutions<br>• IFAC<br>• IOSCO<br>• World Economic Forum | • Accounting professionals<br>• Bank Industry<br>• Investment institutions<br>• Academics |
| Verification Mechanism | Quantitative Performance Indicator-Information can be subjected to examination | Tick Box Approach Lack reporting indicators | Quantitative Performance Indicator-Information can be subjected to examination |
| Country-by-Country Reporting Provision | There are provisions for country-by-country reporting | No | No |
| Sector-Specific Requirements | Yes | No | Yes |

Source: Authors' Elaboration.

In Table 4, this study identifies the influence of each individual issue amplifier in the sustainability reporting regulatory arena, from their perspective and the interest they represent and defend. First, this paper identifiesthat there is diversity in their approach to-mediating and resolving the issues in the arena, which can be linked to the specific interest they aim to protect: "financial/capital market's interest vs. people and the planet's interest". This explains the difference in their core materiality: financial materiality vs. double mate-riality. By definition, whilefinancial materiality focuses on all aspects that are financially significant in decision-making: double materiality is considered as both "financial materi-ality and impact materiality, where impact materiality involves identifying sustainability matters that are material in terms of the impacts of the reporting entity's own operations and its values chain, based on (i) the severity (scale, scope and remediability) and, when appropriate, the likelihood of actual and potential negative impacts on people and the environment; (ii) the scale, scope and likelihood of actual positive impacts on people and

the environment connected with companies' operations and value chains; (iii) the urgency derived from social or environmental public policy goals and planetary boundaries" [81]. Hence, this explains the significant difference in their audience and the nature of the reporting proposed.

In this context, this study argues that there is a connection between the interest of the issue amplifiers and their specific key pressure groups. This perspective is based on the associationof the specific tenet of sustainability reporting that each issue-amplifier is promoting with the professional and technical focus of their key pressure groups. This, however, suggests the possibility of power interplay and lobbying from different pressure groups to influence the shape of the sustainability regulatory arena via the issue amplifiers. For example, Flower [27] questions the controlling power and dominance of the accounting professionals in the governance system of the IIRC, which he regards as "regulatory capture". Complementing this, Reuter and Messner [82] analyzed the responses the IIRC received during the consultation period for the development of the <IRF>. They found that over 32% of the letters received came from accounting professionals, which represents the highest contribution received from a particular stakeholder group. Further, Dumay et al. [83] argue that the influence of the accounting profession on the interests of the IIRC continues to grow to referencethe council being dominated by accounting professionals, particularly from the big four accounting firms (such as the PWC and KPMG). This is also applicable in the case of SASB, whose governance system is more dominated by representatives from the accounting and the banking industry [84]. Therefore, it can be argued that the behavior of the issue amplifiers signals an element of competition, motivated by their need to defend the technical authority and interest of their key pressure group. In this sense, the overarching premise of this study remains the framing activities of the various issuesamplifiedarefar from ensuring harmony in the sustainability reporting arena. This draws us to the implications of the IFRS Foundation and EFRAG's entry into the arena, and the role of the rule enforcer.

## 5. Role of the Rule Enforcer's (European Commission and EFRAG) Position and Influence in the Contested Arena

With the analysis in the previous section, it can be seen that the behavior and influence of the issue amplifiers have increased the confusion and contested issues in the arena. The contested issues now include two major orientations: "financial market and enterprise value creation" vs. "people, organizations and society protection", and "impact disclosure standards" vs. "financial related-sustainability context principles". Following the arena concept, this study identifiesthe European Commission and EFRAGas regulatorsthat can enforce and legitimate any specific orientations for European companies.Although there is a premise that EFRAG is not a standard setter [19]; however, the influence and contributions of EFRAG toward the European Commission's endorsement and acceptance of accounting standards in Europe havebeen documented in prior studies [85,86].Therefore, the works of EFRAG, so far, will be used as the basis to comprehend the direction of the European Commission in this sense.

Following the European Commission's consultation on possible changes to the Directive 2014/95/EU, 588 comment letters were received and 82% of the respondents believed that companies should use common standards to reduce the issue of the lack of comparability. Additionally, 72% of the respondents support that companies should be required to disclose their materiality assessment process [87]. Consequently, EFRAG provided its final report in February 2021, which emphasizes "dynamic materiality"to be of priority in the European Union's sustainability reporting standards-setting.Thisincludes "impact materiality and financial materiality" with a focus on wider stakeholders. The former is defined as "sustainability matters that reflect the reporting entity's significant impact on the environment and people", and the latter is explained as "sustainability matters that create or erode enterprise value and financial material" [81]. In April 2021, the European Commission adopted a proposal for a Corporate SustainabilityReporting

Directive (CSRD), which follows the EFRAG's recommendations and their consultations on Directive 2014/95/EU [88].

The CSRD forms part of the European Green Deal and consists of different measures intended to improve the flow of money toward sustainable activities in the EU. Importantly, other than climate and the environmental objectives of the European Green Deal, it also acknowledgesthat other sustainable development issues should be considered. The main changes proposed in the CSRD include the application of the new rules to large companies (listed or non-listed-removing the 500 employees threshold), SMEs (other than listed micro-enterprises); disclosure of a wide range of sustainability information relevant to business activities, not just environmental factors, but to also include social factors (e.g., gender equality: respect for human rights) and governance factors (business ethics: anti-bribery and corruption) [88].

With the review of the EFRAG's recommendations, it can be argued that the European Commission's adoption of the CSRD represents a declaration of driving towards a greener economy, through the pursuance of a range of simultaneous regulatory interventions. This, however, points to a potential case for debate on whose interest, among the issueamplifiers, will be enforced by the European Commission in pursuing its ambition. Therefore, this takes us to the evaluation of the potential implications and contributions of the IFRS Foundation and EFRAG as the new participants in the arena.

*Intervention of the New Participants in the Arena-Potential Influence and Implications*

Considering the recent drive made by the European Commission to develop separate sustainability reporting standards, its underlying agenda remains connected to fulfilling its ambition of achieving the European Green Deal, thus defending its technical authority. However, stakeholders of the IFRS Foundation specifically asked: to create an international non-financial reporting standards board to resolve the complexity present for investors in assessing the link between non-financial performance and value creation [15]; to ensure comparable and assurable information pertinent to sustainable development and enterprise value creation [9,10]; to build the bridge between the key set of global metrics present among the existing global standard setters [8].In turn, the IFRS Foundation's Trustees published a consultation paper, and associates the case for its intervention with the urgent need to improve comparability and consistency in the arena, and "ensuring the provision of sustainabilityinformation that will be most relevant and useful to investors and other market participants" [89]. Consequently, the Foundation received 577 comment letters and announced that it hada compelling case to intervene and "improve the global consistency and comparability in sustainability reporting" [17].Thus, the Trustees established the Technical Readiness Working Group (TRWG) to do preparatory work towards the establishment of the ISSB and announced its strategic direction. Following this, the strategic direction and the recommendation proposed to the European Commission by the EFRAG's Project Task Force on EU sustainability reporting standards can be reviewed (see Table 5).

In line with Table 5, it can be argued that the interests of the new actors and participants in the arena are contrary tovarious key features of sustainability reporting. In particular, EFRAG's interest is to drive sustainability impact change in a framework of dynamic materiality (by focusing on enterprise value and concurrently addressingthe concerns with the impact on society and people). On the other hand, the IFRS Foundation's interest remains inprotecting investors' interests by focusing on enterprise value creation, while their position on other sustainability reporting issues remains unclear. This points toward the IFRS Foundation's lack of interest in the harmonization of different regulatory framing, as previously proposed. The strategic direction of EFRAG is consistent with the GRI's framing, and the IFRS Foundation aligns with the interests of the SASB and IIRC. Importantly, the proposed consolidation of the VRF into the Foundation suggests that the approach taken by the Foundation is premeditated.

**Table 5.** The strategic direction and influence of the new actors in the sustainability reporting arena.

| New Participants | Audience | Scope and Core Priorities | Approach and Materiality | Pressure Group and Source of Influence |
|---|---|---|---|---|
| **IFRS Foundation-TRWG** | Investors | Primary: Enterprise value-investor's interest and climate related risks<br>Secondary: Other ESG matters | Approach-Financial materiality: "entity's impact on society and the environment, if those impacts could reasonably be expected to affect the entity's cash flow" Information considered material to investors, lenders, and other creditors | IOSCO, IFAC, World Bank, World Economic Forum, ICAEW, Accountancy Europe |
| **EFRAG-PTF** | Multi-stakeholder | All organizations' ESG issues (including its impact via broader value chain).<br>Climate as a key element of the EU's sustainability reporting standards, and adequate coverage of key sustainable themes-ESG<br>External assurance<br>Large companies(listed or not)<br>Integrating the EU's sustainable finance disclosure regulation and Green Taxonomy and guidelines on reporting climate-related information | Dynamic materiality approach: double materiality (society's interest; social and environment protection), and financial returns protection | Legislative support and backing from the European Commission. European national standard setters |

Source: Authors' Elaboration.

Moreover, the analysis reveals that the stakeholders that most pushed for the intervention of the IFRS Foundation, are the same groups that dominated the SASB and IIRC. In agreement, the IOSCO explicitly declares to help the Foundation "enforce the new sustainability standards" [90]. Other groups, particularly in the accounting professions (such as the ICAEW and IFAC), have also emphasized their support in different reports [91,92]. This points toward the argument raised in Adams and Abhayawansa's [6] study: calls for harmonization of sustainability reporting frameworks appear to be driven by a "desire to remove control of sustainability reporting standard-setting away from a multi-stakeholder process". Therefore, the development of different sustainability reporting standards by the Foundation's ISSB may lead to what Flower [27] refers to as "regulatory capture". First, the IOSCO has declared to facilitate the global adoption of the standards within its own technical capacity by mandating them for stock exchange members [9,90]. This may become an avenue to generate an enormous sustainability reporting audit market, particularly for big accounting firms (The Big Four; KPMG, Ernest and Young, Deloitte and PwC).

In agreement, there is historical documentation that the big accounting firms influenced the development of the specific IFRS [93]. Likewise, Wingard and Bosman [94] submit that the IFRS Foundation relies strongly on specific institutions, such as the Big Four (accounting firms), the IOSCO, G8, and the European Commission for funding. In this sense, the premise is that, with the development of the ISSB, there will be at least three sustainability reporting standards to contend with globally: (1) investors and enterprise value creation focus standards from the IFRS Foundation and the VRF (based on financial materiality), (2) impact disclosure and sustainability context standards for all stakeholders by the GRI (based on double materiality), and (3) impact and sustainability disclosure standards for multi-stakeholder specifically for EU companies (based on dynamic materiality approach: double materiality (society's interest; social and environmental protection) and financial returns protection). Therefore, this suggests that there is a risk involving a change in the rules and direction within the arena by the IFRS Foundation, with the potential mobilization of an audience to financial materiality as the core tenet of sustainability reporting. Evidence from prior studies has illustrated that sustainability reporting driven by financial materiality, possesses the potential to have an insignificant and/or negative impact on sustainable development [95–97].

Although EFRAG has the legislative power and the backing of the European Commission, there is a significant concern for the potential for political interference that might occur due to the historical relationship that exists between EFRAG, the European Commission and the IFRS Foundation. Therefore, this demonstrates the various political decisions, power interplay, and influences that may occur behind the scenes.

## 6. Discussion and Conclusions

This study explores how the sustainability reporting terrain has become a contested arena, including the influence and behavior of different actors inshaping the regulation, and how this contributes to the harmonization and future direction of sustainability reporting. Prior studies on the harmonization of sustainability reporting have documented various myths that surround the issues of regulation in this arena and diversity in the materiality perspectives of the various external actors [3,6,26]. However, this study provides different knowledge on the behavior and extent of influence among the key existing external actors, and how they are attempting to influence the rules and shape sustainability reporting regulation, plus what it could mean for the future direction of sustainability reporting.

First, the application of the arena metaphor reveals that sustainability reporting takes place in a complicated environment, with complex interaction and strategy. Expanding this, as seen in the analysis, while rule enforcers are identified as powerful and influential, there are other actors with influence and power, whose specific strategy and interaction havethe potential to change the rules in the arena. Thus, the action and choice of strategy of the various actorsare premeditated and calculated specifically to maintain their influence and relevance, and defend their technical authority within the contested arena. More specifically, the analysis point out that hegemony is perceived as unacceptable among the arena participants, due to the engagement tactics used. In agreement, and following the trend of the development of the various frameworks and standards by the issue amplifiers and the gap they fill at each specific time, this study identifies that they are deliberately exploring individual weaknesses and limitations. Therefore, this suggests that harmonization of sustainability reporting regulation is practically impossible, as the actions and behavior of the issue amplifiers question the ability of any of the actors to renounce their particular perspective and orientation.

Second, the analysis demonstrates the evidence documented in prior studies that there is diversity in the beliefs and interests of the arena participants [25]. It can be observed that the diversity is significantly influenced by the power and professional influence of the specific audience of each participant, especially those the issue amplifiers aim to serve. This is further reinforced bythe positive relationship that exists between the interests of the various actors that dominate the VRF and the interests of the actors that are pushing for the development of the IFRS Foundation's sustainability reporting standards. It is found that, in addition to social influence, power, money, and due process, "collaboration at technical level", "coalition" and frequent consultation with audiences that possess a similar ambition and have significant influence, are tactics used by actors to become more powerful and help them pursue their objective and interest in an arena. For instance, the development of the ISSB by the IFRS Foundation and its consolidation of the VRF despite the existence of the GRI's GSSB and the ongoing work of EFRAG/European Commission, suggests that the former has the potential to hijack and subvert the sustainability agenda for enterprise value. The IFRS Foundation ignores how enterprise impacts the globe and the social and ecological thresholds that define sustainability, which is consistent with the characteristics of the arena concept, which denote thatseveral participants with similar interests may "join forces to change rules in the arena" [24].

This, however, reinforces the case that sustainability reporting regulation is still far away from harmonization with various contested issues still yet to be resolved. First, the behaviors among the actors in the sustainability reporting arena suggest there is still contestation on whether there should be an artificial boundary between "how entities are affected and depend on sustainability relevant external factors" and "how entities

impact them and the world in turn". Second, there are different notions of materiality (financial materiality, double materiality and dynamic materiality) associated with different interpretations, which further increases the level of misunderstandingand confusion in the arena. Therefore, with the social and ecological thresholds that define sustainability, it can be argued that the possibility of driving future sustainability reporting regulation toward ensuring companies disclose their true impacts, remains unlikely.

Although the analysis reveals that theGRI remains the dominant global institution for sustainability reporting. Its revised standards offer a clear modular set to provide an inclusive picture of material topics (beyond financial materiality), with the implementation of specific standard setting practice lingo. This demonstrates the GRI efforts in preserving the core tenet of sustainability reporting and ensuring that multi-stakeholders hold organizations to account for how their decisions and activities impact their welfare and the planet. However, this study argues that the standards are based on "sustainability context impact" and not "context based approach", because the GSSB "basis for conclusions" stated: "the GRI standards do not set allocations goals, targets, thresholds, or any other benchmarks for sustainable (good) and unsustainable (bad) performance" [98]. Therefore, the question that remains is whether impact standards without commitment to sustainability performance assessment are adequate to preserve the future of sustainability reporting. Consequently, it can be argued that the behavior and interactions of the various actors aregradually changing the rules and focus in the arena. However, the arena concept suggests that whether an actor fails or succeeds, depends on the amount of influence they can exercise on the resulting policy or decision [24]. This suggests there is an urgent need for the most powerful and influential actors in the arena (political institutions and rule enforcers) to decide and reclassify the rules upholding the arena.

Nonetheless, like other studies, this research has certain limitations. While there is information available on the different actors contributing and proposing ideas toward the future direction of the sustainability reporting regulatory arena, this study only focuses on a few actors and examined limited information and public documents. Thus, itcan be identifiedthat this perspective may exclude other important insights from other participants in the arena, such as the TCFD, CDSB, different companies, and campaigning NGOs. Additionally, there are more arguments and political debates going on behind the scenes that havenot been studied. However, it can be argued that the chronicle within the sustainability reporting regulatory arena does not stop here, due to the different potential interactions and political decisions that can be made and influenced by various powerful and influential actors in the arena. Future research can continue the investigation through a different lens, such as interviewing different actors, particularly the key members of the GRI, VRF, EFRAG and the IFRS Foundation to comprehend what may have led to their choice of strategic decision and explore this more critically.

**Author Contributions:** Writing–original draft and revision improvements, H.A.; review & editing, R.R. and G.R. All authors have read and agreed to the published version of the manuscript.

**Funding:** This research received no external funding, but received 100% discount voucher for the article publication charges as an invited paper for the special issue (provided by sustainability journal).

**Institutional Review Board Statement:** Not applicable.

**Informed Consent Statement:** Not applicable.

**Data Availability Statement:** The datasets used during the current study are publicly available resources/information gathered by the first author, and their sources are adequately referenced.

**Acknowledgments:** We thank the three anonymous reviewers for providing valuable suggestions and comments that improved the quality of the paper. Also, we thank and appreciate the support of the academic editor of the special issue, Guler Aras.

**Conflicts of Interest:** The authors declare no conflict of interest.

## Abbreviations

| Abbreviation | Meaning |
|---|---|
| GRI | Global Reporting Initiative |
| GSSB | Global Sustainability Standard Board |
| IIRC | International Integrated Reporting Council |
| <IRF> | Integrated Reporting Framework |
| SASB | Sustainability Accounting Standard Board |
| CDP | Carbon Disclosure Project |
| CDSB | Climate Disclosure Standards Board (CDSB) |
| TCFD | Task Force on Climate-related Financial Disclosure |
| IOSCO | International Organizations of Securities Commission |
| ICAEW | Institute of Chartered Accountants in England and Wales |
| IFAC | International Federation of Accountants |
| EFRAG | European Financial Reporting Advisory Group |
| CSRD | Corporate Sustainability Reporting Directive |
| ESG | Environment, Social and Governance |
| PTF | Project Task Force |
| IFRS Foundation | International Financial Reporting Standard Foundation |
| ISSB | International Sustainability Standard Board |
| TRWG | Technical Readiness Working Group |
| VRF | Value Reporting Foundation |
| SLR | Systematic Literature Review |

## Appendix A

**Table A1.** Summary of the articles selected and their key findings.

| Author(s)/Year | External Actors and Framework Studied | Objective/Question | Key Findings |
|---|---|---|---|
| Brown et al. (2009a) | GRI | Examined GRI's organizational field and impact as a mobilizing agent for many societal actors | GRI has been a successful institutionalization project by many measures, but its emerging institutional logic reflects only some of its intended constituencies, namely MNCs, and financial institutions, and international business management consultancies and accountancies |
| Brown et al. (2009b) | GRI and its framework | Explore the strategies, framing ideas and origin of the vision of the GRI in creating a new global institution for sustainability reporting | The GRI has the potential to foster social change within the context of SR, however, there are other inherent challenges in relation to its history and principles. The resolution of these challenges would determine its future shape. |
| Flower (2015) | IIRC and <IR> 2013 Framework | The history, and objectives of the IIRC and its Framework | Concluded that in the IRF, the IIRC has abandoned sustainability accounting due to its value for investors and not for society. Additionally, the IIRC will have limited impact on corporate reporting practice, because of lack of force in its framework |
| Levy et al. (2010) | GRI | Investigate the GRI's strategy, success and problems | GRI has fallen short of the aspirations of its founders to use disclosure to empower non-governmental organizations (NGOs). Its trajectory reflects the power relations between members of the field, their strategic choices and compromises, their ability to mobilize alliances, and constraints imposed by the broader institutions of financial and capital markets. |
| Moneva et al. (2006) | GRI | What is the potential impact of the GRI guidelines for sustainability development and possible gaps within the guidelines? | The GRI guidelines remain an administrative reform that is yet sufficient to encourage accountability |

**Table A1.** *Cont.*

| Author(s)/Year | External Actors and Framework Studied | Objective/Question | Key Findings |
|---|---|---|---|
| Dumay et al. (2017) | <IR> 2013 Framework | What are the enablers, incentives and barriers in implementing the IRF? | The lack of prescription and flexibility in actual disclosures of the IRF could improve it adoption for compliance. However, it diverse way to be enacted pose numerous empirical and theoretical challenges for academics |
| Adams (2015) | IIRC and <IR> 2013 Framework | How IR can change the thinking of corporate actors leading to further integration of sustainability impacts and actions into corporate strategic decision making and planning? | There is distinction between sustainability reporting and IR. Whilst the idea o IR will evolve, the impact of the reporting practice of the IIRC and IRF will depend on those critical of the status quo. |
| Tweedie and Martinove Bennie (2015) | IIRC | What are IIRC'sdistinctive philosophy, objectives and implications of their reporting approach for sustainability? | IR moves away from the key tenets of environmental and social reporting, however, it has a potential to shift financial capital to long term investment horizon from the current short term. |
| Dumay et al. (2010) | GRI guidelines | Provides a critique of the GRI guidelines, and examines their applicability to public and third sector organizations | The guidelines promote a 'managerialist' approach to sustainability rather than an ecological and eco-justice informed approach, potentially causing them to fall into an evaluator trap |
| Reuter and Messner (2015) | IIRC | Examined the formal participation in the early phase of the IIRC's standard-setting | Observe active lobbying by sustainability service firms and professional bodies, which tend to take critical position vis-a-vis the discussion paper's emphasis on investor needs and shareholder value creation. |
| Brown and Dillard (2014) | IIRC | Critically reflect on the IIRC's advocacy of a business case approach to integrated reporting as an innovation that can contribute to sustainability reporting | Integrated reporting, as conceived by the IIRC, provides a very limited and one-sided approach to assessing and reporting on sustainability issues |
| Hedberg and Malmborg (2003) | GRI Framework | Explores why companies have chosen to use the GRI guidelines and how this has affected corporate social responsibility and environmental management. | The main reason for use of the GRI guidelines is an expectation of increasing credibility of the CSR, but also that it provides a template for how to design a report. |
| Wagner and Seele (2017) | GRI-G3 and G4 | Investigate the differences between G3 and G4 and their application | There is lack of understanding and guidance on how to apply GRI principles |
| Safari and Areeb (2020) | GRI-Principles for Defining Report Quality | Analyze how sustainability report preparers perceive the GRI-principles for defining report quality and explore opportunities, challenges and influential factors that reports preparers experience in the application of these principles | Under-developed reporting systems, along with time and cost constraints, have served as prominent barriers to efficient practicalization of the principles for defining report quality |
| Isaksson and Steimle (2009) | GRI guidelines | Explore the extent of the GRI guidelines offer clear and relevant sustainability information disclosure | The current GRI guidelines are not sufficient to make sustainability reporting for the cement industry relevant and clear |

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
