# Peer review of "Harmonization of Sustainability Reporting Regulation: Analysis of a Contested Arena"

_sustainability, doi:10.3390/su14095517_

Round 1
Reviewer 1 Report
Thank you for the opportunity to review this manuscript. The paper examines the Non-Financial Reporting (NFR) regulatory market. It seeks to explore the influence of the various external actors (GRI, SASB, IIRC, EC, EFRAG and the IFRS Foundation) on the changing NFR landscape. More precisely, it aims to investigate "how they compete to be the most influential shaper of NFR" (line 814)).
The debate around the regulation of NFR among practitioners and academia members intensified after the IFRS Foundation's decision to get involved in this process. Therefore, I find this paper very relevant to everyone interested in the future of NFR. In my opinion, it has the potential to contribute new insights into the literature on NFR and has practical implications. However, some minor concerns have been identified, as detailed in the specific sections of this review. I hope that these comments will provide some useful feedback, and I wish the authors the best of luck with further development of this paper.
The abstract and the introduction
The abstract is well written and it includes all the necessary information.
The motivation is presented in the first part of the introduction and followed by a statement about the research aim ("(…) this study seeks to explore the role of various external actors, including their contestation for dominance and influence, and the factors influencing the competition amongst them to make sense of their impact for the future direction of the NFR regulatory market").
The information on the research method (including systematic literature review (SLR)) and the main findings could be presented in the introduction section as well. This would clarify the study's contribution to theory and practice. The introduction section ends with information about how the paper is organized. In the interest of consistency, I would suggest referring to all section numbers 1-5 or using no numbers at all.
The theoretical background
Renn’s (1992) arena approach seems a helpful and appropriate theoretical underpinning. You also refer to the legitimacy concept in the paper (e.g., "Following our analysis, the controversy at stake in this contested arena can be conceptualised as a struggle for legitimacy amongst the participants, particularly the need to defend their individual technical authority" (lines 816-817)). Would you not agree that this concept might be explained in more detail and also used as a theoretical background?
You claim that „The rule enforcers mostly gain their power legislatively from political institutions” (line 144-145). Why is there no arrow between the boxes representing them (rule enforcers and political institutions) in Figure 1?
In my opinion, Figure 2 is a bit confusing. Why are EFRAG and IFRS Foundation outside the box? Why do the circles representing SASB and IIRC overlap with the GRI circle and IIRC and SASB are separated?
EU Commission and EFRAG are both rule enforcers, yet their visualization is different.
I wonder why you have not decided to replicate Figure 1, but with the names of the actors you focus on in your paper. I think it would be clearer to the readers.
Who are the political institutions in the NFR regulatory market?
The research method
I would suggest using the term method instead of methodology.
How do you define method and how do you define methodology? Unfortunately, the terms method and methodology are often used interchangeably. I know that methodology is repeatedly used in the context of the research approach, but in fact, "methods” are the means whereby one collects and analyses data. Methodology refers to the philosophical issues which underlie those methods (see Guthrie et al., 2004, p. 417; de Villiers and Dumay 2013, p. 893).
You present the SLR as your research approach, and I understand that it was used by you to “examine the extent at which NFR regulatory market has become a contested arena, including how the key external actors are competing for dominance and influence” (lines 228-229). However, it is not clear to me how exactly these 45 articles were used in your study. Do you cite all these studies in the sections that follow (4,5,6)? I expect that their analysis was useful to you, but I think that there should be a stronger connection between the method you rely on (SLR) and describe in section 3 and your findings presented in the sections that follow (4,5,6). What exactly is the purpose of identifying the country of analysis or theory used? You explain that, at the final stage of the analysis, you focus on specific information (year, focus, timeframe, findings (lines 314-315)), and that you map the development of the regulations with the studies.
Table 1 presents the “documents from key stakeholders/standard-setters examined in this study”. Similar information about the examined papers is missing. I assume that providing them in an appendix with all the relevant information you relied on might not be feasible due to the limitations of the paper’s length.
Figure 1 is unreadable. I see eight black boxes and the text does not seem to fit in other boxes properly. Are the thematic areas mentioned in table 3 the same as the keywords described in lines 265-267?
Sections 4, 5 and 6
Figure 4 is very important as it organizes the text flow in the following sections. The text in some boxes is not fully visible. Maybe you could consider transforming this figure into a table with information about the date and the event description.
In figure 4, you mention CDP and CDSB for the first time. In fact, together with GRI, IIRC, SASB, they formed the so-called “big five” – the most influential institutions when it comes to the NFR regulation.
Are CDP and CDSB issue amplifiers? The title of Figure 4 suggests they are. Maybe they should also be mentioned in the main body of the paper? How about TCFD?
In table 4 you present Big 4 as “key stakeholders involved (pressure groups)” under IIRC and “accounting firms” under SASB. My understanding is that the accounting profession in general was and still is involved in the IIRC operation, not only Big 4. In the paper you say that "Complementing this view, Reuter and Messner (2015, p.375) analysed the responses the IIRC received during the consultation period for the development of the <IRF>. They found out that over 32% of the letters received came from the accounting professionals, which represent the highest contribution received from a particular stakeholder group" (lines 689-691). Were only Big 4 companies the respondents? How is the situation different with the SASB?
In section 6 you introduce the term "NFRD 2014/95". I would suggest using the wording "Directive 2014/95/EU".
Discussion and Conclusion
The discussion and conclusion section is well written. It ties together the other elements of the paper adequately. I only suggest including a more straightforward reference to Renn’s (1992) theory in this section.
I think that you could be more specific when it comes to explaining how your study contributes to the literature. Would it be possible to state which of the studies you revised during SLR are challenged or reinforced by your findings (e.g. that hegemony is perceived unacceptable amongst the arena participants and competition is the underlying trend when it comes to the development of the various guidelines by the issue-amplifiers)?
You argue that competing motivations and" 'coalition', 'technical collaboration' and 'frequent consultations with the key pressure groups' remain the significant engagement tactics use (used?) by various stakeholders" (line 847-849). So what? How would this affect the future NFR reporting landscape?
The practical implications for NFR policymakers are sufficiently demonstrated. However, maybe you could indicate your limitations and present further research more clearly.
The quality of English is satisfactory. I have only a few minor comments:
- line 35 – the abbreviation SR should be explained;
- line 54 - EC's Directive 54 2014/95/EU/NFRD – I would suggest referring to the Directive as "Directive 2014/95/EU;
- the abbreviation NFDR is used 4 times in the text, but I think that it is not necessary; “NFRD 2014/95/EU” – is just too much and it is uncommon to refer to the Directive this way;
- line 58 – improve complexity or reduce complexity?
- 4, Figure 1 – is a hyphen needed in "stake-holders"?
- line 515 – The title of subsection 5.1.2 should be in a separate line;
- please correct the way you use the capital letters in the text – sometimes not necessarily (e.g., in table 5 "Double Materiality-Both Financial materiality (…) “);
- line 865 – the abbreviation FR should be explained;
- line 868 - I would rather say NFR assurance market than NFR audit market;
- the way you provide the references is not in line with the journal's instruction for authors (references must be numbered in order of appearance in the text).
Author Response
Reviewer 1:
The abstract and the introduction
The abstract is well written and it includes all the necessary information.
The motivation is presented in the first part of the introduction and followed by a statement about the research aim ("(…) this study seeks to explore the role of various external actors, including their contestation for dominance and influence, and the factors influencing the competition amongst them to make sense of their impact for the future direction of the NFR regulatory market"). The information on the research method (including systematic literature review (SLR)) and the main findings could be presented in the introduction section as well. This would clarify the study's contribution to theory and practice. The introduction section ends with information about how the paper is organized. In the interest of consistency, I would suggest referring to all section numbers 1-5 or using no numbers at all.
Authors’ Response: We have now provided additional information in the introduction section relevant to the research method and the key findings of this study. Also, the remaining sections of the study are provided at the end of the introduction in similar vein to ensure consistency as the reviewer suggested.
The theoretical background
Renn’s (1992) arena approach seems a helpful and appropriate theoretical underpinning. You also refer to the legitimacy concept in the paper (e.g., "Following our analysis, the controversy at stake in this contested arena can be conceptualised as a struggle for legitimacy amongst the participants, particularly the need to defend their individual technical authority" (lines 816-817)). Would you not agree that this concept might be explained in more detail and also used as a theoretical background? You claim that „The rule enforcers mostly gain their power legislatively from political institutions” (line 144-145). Why is there no arrow between the boxes representing them (rule enforcers and political institutions) in Figure 1?
In my opinion, Figure 2 is a bit confusing. Why are EFRAG and IFRS Foundation outside the box? Why do the circles representing SASB and IIRC overlap with the GRI circle and IIRC and SASB are separated? EU Commission and EFRAG are both rule enforcers, yet their visualization is different. I wonder why you have not decided to replicate Figure 1, but with the names of the actors you focus on in your paper. I think it would be clearer to the readers. Who are the political institutions in the NFR regulatory market?
Authors’ Response: We agree that the reviewer raised a valid concern. Following the reviewer’s additional comment in the ‘discussion and conclusion’ section, we thought carefully about how we mobilized the theory. (see below):
“You argue that competing motivations and coalition, technical collaboration and frequent consultations with the key pressure groups remains the significant engagement tactics used by various stakeholders….So what? How would this affect the future NFR reporting landscape”.
The reviewer’s comment above helped and informed us to reflect on the framing of the research objective and question, including the practical implication of the research. Therefore, following the previously investigated objective, the revised version has now taken the reviewer’s comment into consideration and now explore the behavior and influence of the key external actors within the arena and how these contribute to the future direction and harmonization of sustainability reporting regulation. Further, the arena concept is now re-mobilized as proposed by Renn (1992) and further developed on by Georgakopoulous and Thomson (2008), which has now enhanced the findings and practical implications of the study. In fact, the position of the EFRAG, IFRS Foundation and other relevant actors, including political institutions are now explained clearly as the reviewer suggested.
Additionally, in response to another concern the reviewer raised regarding our claim that “the rule enforces mostly gain their power legislatively from political institution”, we would like to mention that this claim is consistent with the Renn’s (1992) explanation of the arena concept. Renn (1992, p.183) provided an arrow between the boxes representing the rule enforcer and political institution as the reviewer also thought. However, the graphical representation used in our paper is the one drawn up by Georgakopoulous and Thomson (2008, p.1120), which is without the arrow. Nonetheless, the interpretation of the relationship that exists between political institution and rule enforcer is still the same.
The research method
I would suggest using the term method instead of methodology. How do you define method and how do you define methodology? Unfortunately, the terms method and methodology are often used interchangeably. I know that methodology is repeatedly used in the context of the research approach, but in fact, "methods” are the means whereby one collects and analyses data. Methodology refers to the philosophical issues which underlie those methods (see Guthrie et al., 2004, p. 417; de Villiers and Dumay 2013, p. 893).
You present the SLR as your research approach, and I understand that it was used by you to “examine the extent at which NFR regulatory market has become a contested arena, including how the key external actors are competing for dominance and influence” (lines 228-229). However, it is not clear to me how exactly these 45 articles were used in your study. Do you cite all these studies in the sections that follow (4,5,6)? I expect that their analysis was useful to you, but I think that there should be a stronger connection between the method you rely on (SLR) and describe in section 3 and your findings presented in the sections that follow (4,5,6). What exactly is the purpose of identifying the country of analysis or theory used? You explain that, at the final stage of the analysis, you focus on specific information (year, focus, timeframe, findings (lines 314-315)), and that you map the development of the regulations with the studies.
Table 1 presents the “documents from key stakeholders/standard-setters examined in this study”. Similar information about the examined papers is missing. I assume that providing them in an appendix with all the relevant information you relied on might not be feasible due to the limitations of the paper’s length.
Figure 1 is unreadable. I see eight black boxes and the text does not seem to fit in other boxes properly. Are the thematic areas mentioned in table 3 the same as the keywords described in lines 265-267?
Authors’ Response: As the reviewer suggested, we have now used the term ‘method’. Further, the systematic literature review (SLR) is used specifically to support the understanding of the institutional strategy and behavior of the external actors (GRI, SASB and IIRC), and the implication of their respective guidelines, frameworks, and standards. The search for the academic literature covered two thematic areas, which include the articles that explored:
- the background of any of the selected external actors (GRI, SASB and IIRC) and concerned with the practice and implication of their guidelines and frameworks and standards issued.
- the approach used by any of the selected external actors in mobilizing provisions and regulating sustainability reporting arena.
The SLR process we used is now provided in Figure 2. This helped us to identify the key articles that are relevant to our study. The selected articles are cited and their analysis were used in our study, particularly in section 4 to corroborate the understanding of the strategy and influence of the external actors (GRI, SASB and IIRC), which help to makes sense of their behavior and how it is affecting the shape of the sustainability reporting regulatory arena. Further, additional information regarding how the documents and articles selected are used is now provided in the method section of the paper. Likewise, the list of the articles used is now provided as appendix for a sample (because of the length of the paper). Also, the eight black boxes that show the process of the systematic literature review has been adjusted and now readable (see Figure 2).
Sections 4, 5 and 6
Figure 4 is very important as it organizes the text flow in the following sections. The text in some boxes is not fully visible. Maybe you could consider transforming this figure into a table with information about the date and the event description. In figure 4, you mention CDP and CDSB for the first time. In fact, together with GRI, IIRC, SASB, they formed the so-called “big five” – the most influential institutions when it comes to the NFR regulation.
Are CDP and CDSB issue amplifiers? The title of Figure 4 suggests they are. Maybe they should also be mentioned in the main body of the paper? How about TCFD?
In table 4 you present Big 4 as “key stakeholders involved (pressure groups)” under IIRC and “accounting firms” under SASB. My understanding is that the accounting profession in general was and still is involved in the IIRC operation, not only Big 4. In the paper you say that "Complementing this view, Reuter and Messner (2015, p.375) analysed the responses the IIRC received during the consultation period for the development of the <IRF>. They found out that over 32% of the letters received came from the accounting professionals, which represent the highest contribution received from a particular stakeholder group" (lines 689-691). Were only Big 4 companies the respondents? How is the situation different with the SASB? In section 6 you introduce the term "NFRD 2014/95". I would suggest using the wording "Directive 2014/95/EU".
Authors’ Response: Thank you for your comments. Figure 4 is now presented as table 3 to enhance its readability and visibility as the reviewer suggested. Further, while we agree with the reviewer that CDSB and TCFD could also form part of the issue amplifiers, our study focused on GRI, IIRC and SASB because of two different reasons:
- While CDSB and TCFD focus on providing useful metrics for companies to promote and advance climate change-related disclosure in mainstream reports, different reports and studies, such as KPMG (2020) as identify the three institutions (GRI, SASB and IIRC) as the ones addressing various key elements involve in sustainability reporting.
- Due to the limitation of words and length of the paper, we decided to focus on the three institutions, including the EFRAG, IFRS Foundation and European Commission.
However, while the above points remain in mind, we have identified this as part of the limitations of the paper and also form part of our suggestion for the future study. Further, another concern mentioned by the reviewer that is related to Reuter and Messner (2015, p.375) is now addressed by demonstrating the contribution of accounting professionals in the development of the SASB. Finally, the term “NFRD 2014/95” is now changed to the word “Directive 2014/95/EU” throughout the paper as suggested by the reviewer.
Discussion and Conclusion
The discussion and conclusion section is well written. It ties together the other elements of the paper adequately. I only suggest including a more straightforward reference to Renn’s (1992) theory in this section. I think that you could be more specific when it comes to explaining how your study contributes to the literature. Would it be possible to state which of the studies you revised during SLR are challenged or reinforced by your findings (e.g. that hegemony is perceived unacceptable amongst the arena participants and competition is the underlying trend when it comes to the development of the various guidelines by the issue-amplifiers)?
You argue that competing motivations and" 'coalition', 'technical collaboration' and 'frequent consultations with the key pressure groups' remain the significant engagement tactics use (used?) by various stakeholders" (line 847-849). So what? How would this affect the future NFR reporting landscape?
The practical implications for NFR policymakers are sufficiently demonstrated. However, maybe you could indicate your limitations and present further research more clearly.
Authors’ Response: Following the re-mobilization of the Renn’s (1992) theory as discussed above (our response under the theoretical background), we have now explained specifically how our study contributes to the literature and practice. Importantly, we have provided a straightforward reference to Renn’s (1992) theory as suggested by the reviewer, which can be found in the second paragraph of the ‘discussion and conclusion section’. Further, the limitations of the study are now highlighted clearly and suggestion for future study is now provided. The reviewer will see this in the last paragraph of the paper.
The quality of English is satisfactory. I have only a few minor comments:
- line 35 – the abbreviation SR should be explained;
Authors’ Response:
- line 54 - EC's Directive 54 2014/95/EU/NFRD – I would suggest referring to the Directive as "Directive 2014/95/EU;
- the abbreviation NFDR is used 4 times in the text, but I think that it is not necessary; “NFRD 2014/95/EU” – is just too much and it is uncommon to refer to the Directive this way;
- line 58 – improve complexity or reduce complexity?
- 4, Figure 1 – is a hyphen needed in "stake-holders"?
- line 515 – The title of subsection 5.1.2 should be in a separate line;
- please correct the way you use the capital letters in the text – sometimes not necessarily (e.g., in table 5 "Double Materiality-Both Financial materiality (…) “);
- line 865 – the abbreviation FR should be explained;
- line 868 - I would rather say NFR assurance market than NFR audit market;
- the way you provide the references is not in line with the journal's instruction for authors (references must be numbered in order of appearance in the text).
Authors’ Response: The abbreviation SR means sustainability reporting and it is now used and written in full throughout the paper. Also, the abbreviation EC’s Directive 2014/95/EU/NFRD is now referred to as Directive 2014/95/EU, and the abbreviation NFDR has been removed as the reviewer suggested. Further, the abbreviation FR means financial reporting and is now explained in the paper, and the NFR assurance market is now referred to as sustainability reporting audit market. The English style of the paper has been professional edited and formatted in line with the journal’s guidelines. Therefore, all other issues the reviewer raised that are related to the English style (such as the “Double materiality-Both Financial materiality, stake-holders”) have all been corrected as the reviewer suggested. Also, the list of the abbreviations used and their meaning is provided as appendix.

Reviewer 2 Report
The aim of this paper is to investigate the influence of the various external actors, such as the GRI, SASB, IIRC, and the European Commission, including the recent intervention of the EFRAG and the IFRS 11 Foundation in the harmonization of the NFR regulation. The paper contains a very valuable research and deals with a relevant topic which justify publication. Context of the study is unique and well explained.
Review of existing literature is adequate. Information regarding harmonisation of NFRs across countries is closely linked with the literature on theoretical foundation. Additionally, there are some parts missing in Figure 3. I could not understand the systematic literature review-research process. The author/authors have to reorganize that figure and provide detailed information related with that part of the study.
These implications consistent with the findings and conclusions of the paper. A clear contribution to practice is recorded. There may potentially be a significant impact on public policy.
Author Response
Reviewer 2
The aim of this paper is to investigate the influence of the various external actors, such as the GRI, SASB, IIRC, and the European Commission, including the recent intervention of the EFRAG and the IFRS 11 Foundation in the harmonization of the NFR regulation. The paper contains a very valuable research and deals with a relevant topic which justifies publication. Context of the study is unique and well explained.
Review of existing literature is adequate. Information regarding harmonisation of NFRs across countries is closely linked with the literature on theoretical foundation. Additionally, there are some parts missing in Figure 3. I could not understand the systematic literature review-research process. The author/authors have to reorganize that figure and provide detailed information related with that part of the study. These implications are consistent with the findings and conclusions of the paper. A clear contribution to practice is recorded. There may potentially be a significant impact on public policy.
Authors’ Response: Thank you for the comment. However, following the concern raised in relation to the method, the systematic literature review-research process has been adjusted and is now readable. Also, the information in figure 3 is now transferred into table 3 to enhance its visibility as the reviewer suggested.
Reviewer 3
The article deals with the Non-Financial Reporting (NFR) regulatory market as a contested arena. It is a very interesting and current topic considering issues of growing interest. The manuscript is potentially relevant for the field.
Nevertheless, some aspects concern me:
- I am not sure that the title represents the manuscript's content properly. The authors present the NFR market as a contested arena where the actors play to win the competition. Moreover, the differences between the participants and their outputs are highlighted many times. Finally, this concept is not recalled in conclusion, not even to deny its existence.
- The methodology is not clear.
In the introduction, line 84 reports: “We take a historical perspective”, line 623 says: “Following the historical analysis in the previous section”, the line 822 writes: “Following our historical and documentary analysis”, whereas section 3 is dedicated to the description of the phases of a systematic literature review. I do not understand the difference between a historical analysis, a documentary analysis, and a literature review focused on the evolution of the NFR regulatory market (that is what your article is, in my opinion). Furthermore, the authors should explain how the documents of table 1 were examined. Finally, the link between the methodology and the considerations in section 4 and the following is unclear.
Authors’ Response: Thank you for the constructive comments. We have now redefined and explained clearly the uncertainties and conditions that support the sustainability reporting field as a contested arena in relation to Renn’s (1992) arena concept. This can be seen in section 2.1 where we discuss “the application of the arena concept in sustainability reporting field”. Consequently, through the aid of arena concept, we have now recalled the contestation in the arena and explained what this could mean for the future direction and harmonization of sustainability reporting regulation as the reviewer suggested.
Further, the method of the paper is now adjusted. The paper draws its data through documentary analysis and systematic literature review. First, for the documentary analysis, public releases of the various actors pertinent to this study have been used to examine the extent at which the sustainability reporting regulatory sphere has become a contested arena, including the behavior of these actors and how they are influencing the shape of sustainability reporting regulation. The study relies on these documents because they reflect the conventional ways the various actors are thinking, and reinforce the pattern in which they are contributing to the regulation in the arena. This helps us to understand the development and changes in their various guidelines, frameworks and standards, to make sense of their behaviour and influence in shaping the regulation within the sustainability reporting arena.
Further, the systematic literature review (SLR) is used specifically to support the understanding of the institutional strategy and behavior of the external actors (GRI, SASB and IIRC), and the implication of their respective guidelines, frameworks and standards. The search for the academic literature covered two thematic areas, which include articles that explored:
- the background of any of the selected external actors (GRI, SASB and IIRC) and concerned with the practice and implication of their guidelines and frameworks and policies issues.
- the approach used by any of the selected external actors in mobilizing provisions and regulating sustainability reporting arena.
The SLR process we used is now provided in Figure 2. This helped us to identify the key articles that are relevant to our study. The selected articles are cited and their analysis were used in our study, particularly in section 4 to corroborate the understanding of the strategy and influence of the external actors (GRI, SASB and IIRC), which help to make sense of their behavior and how it is affecting the shape of the sustainability reporting regulatory arena. Further, additional information regarding how the documents and articles selected are used is now provided in the method section of the paper. Likewise, the list of the articles used is now provided as appendix for a sample (because of the length of the paper). Also, the eight black boxes that show the process of the systematic literature review has been adjusted and now readable (see Figure 2).
- The authors should explain the meaning of “ideology” (line 679) in the context of this work.
- The figures are sometimes not precise. According to the authors, figure 2 represents the NFR regulation market as a contested arena, but it is so different from figure 1 (source Renn, 1992) that it is difficult to interpret. Figure 3 should show the number of papers selected at each phase.
- subsection 6.1 (or only its title?) is missing
- In the section “Discussion and Conclusion”, I did not find the answer to the second research question: the authors seem more focused on the behavior of actors in the contested arena rather than on “the factors influencing the competition in the NFR regulation market”.
Authors’ Response: The meaning of “ideology” as used in the paper implies the specific interest each of the external actors is pursuing in the arena. We have redefined this in line with the research objective as the reviewer suggested. Also, all the figures that are not readable previously have been transferred into tables, and can now be read clearly. In addition, the arena concept has been re-mobilized, therefore, the position and definition of each actors in the arena has been re-defined in line with the explanation of arena concept by Rena (1992). This can be seen in section 2. Finally, the paper now focuses on the behavior and influence of the various external actors within the arena, and how this contributes to the future direction and harmonization of sustainability reporting regulation. This is in line with the reviewers’ comments for us to demonstrate and clearly show the practical implication of the paper and it connection with the theoretical framework.
Reviewer 3 Report
The article deals with the Non-Financial Reporting (NFR)regulatory market as a contested arena. It is a very interesting and current topic considering issues of growing interest. The manuscript is potentially relevant for the field.
Nevertheless, some aspects concern me:
- I am not sure that the title represents the manuscript's content properly. The authors present the NFR market as a contested arena where the actors play to win the competition. Moreover, the differences between the participants and their outputs are highlighted many times. Finally, this concept is not recalled in conclusion, not even to deny its existence.
- The methodology is not clear.
In the introduction, line 84 reports: “We take a historical perspective”, line 623 says: “Following the historical analysis in the previous section”, the line 822 writes: “Following our historical and documentary analysis”, whereas section 3 is dedicated to the description of the phases of a systematic literature review. I do not understand the difference between a historical analysis, a documentary analysis, and a literature review focused on the evolution of the NFR regulatory market (that is what your article is, in my opinion).
Furthermore, the authors should explain how the documents of table 1 were examined.
Finally, the link between the methodology and the considerations in section 4 and the following is unclear.
- The authors should explain the meaning of “ideology” (line 679) in the context of this work.
- The figures are sometimes not precise. According to the authors, figure 2 represents the NFR regulation market as a contested arena, but it is so different from figure 1 (source Renn, 1992) that it is difficult to interpret. Figure 3 should show the number of papers selected at each phase.
- subsection 6.1 (or only its title?) is missing
- In the section “Discussion and Conclusion”, I did not find the answer to the second research question: the authors seem more focused on the behavior of actors in the contested arena rather than on “the factors influencing the competition in the NFR regulation market”.
Author Response
Reviewer 3
The article deals with the Non-Financial Reporting (NFR) regulatory market as a contested arena. It is a very interesting and current topic considering issues of growing interest. The manuscript is potentially relevant for the field.
Nevertheless, some aspects concern me:
- I am not sure that the title represents the manuscript's content properly. The authors present the NFR market as a contested arena where the actors play to win the competition. Moreover, the differences between the participants and their outputs are highlighted many times. Finally, this concept is not recalled in conclusion, not even to deny its existence.
- The methodology is not clear.
In the introduction, line 84 reports: “We take a historical perspective”, line 623 says: “Following the historical analysis in the previous section”, the line 822 writes: “Following our historical and documentary analysis”, whereas section 3 is dedicated to the description of the phases of a systematic literature review. I do not understand the difference between a historical analysis, a documentary analysis, and a literature review focused on the evolution of the NFR regulatory market (that is what your article is, in my opinion). Furthermore, the authors should explain how the documents of table 1 were examined. Finally, the link between the methodology and the considerations in section 4 and the following is unclear.
Authors’ Response: Thank you for the constructive comments. We have now redefined and explained clearly the uncertainties and conditions that support the sustainability reporting field as a contested arena in relation to Renn’s (1992) arena concept. This can be seen in section 2.1 where we discuss “the application of the arena concept in sustainability reporting field”. Consequently, through the aid of arena concept, we have now recalled the contestation in the arena and explained what this could mean for the future direction and harmonization of sustainability reporting regulation as the reviewer suggested.
Further, the method of the paper is now adjusted. The paper draws its data through documentary analysis and systematic literature review. First, for the documentary analysis, public releases of the various actors pertinent to this study have been used to examine the extent at which the sustainability reporting regulatory sphere has become a contested arena, including the behavior of these actors and how they are influencing the shape of sustainability reporting regulation. The study relies on these documents because they reflect the conventional ways the various actors are thinking, and reinforce the pattern in which they are contributing to the regulation in the arena. This helps us to understand the development and changes in their various guidelines, frameworks and standards, to make sense of their behaviour and influence in shaping the regulation within the sustainability reporting arena.
Further, the systematic literature review (SLR) is used specifically to support the understanding of the institutional strategy and behavior of the external actors (GRI, SASB and IIRC), and the implication of their respective guidelines, frameworks and standards. The search for the academic literature covered two thematic areas, which include articles that explored:
- the background of any of the selected external actors (GRI, SASB and IIRC) and concerned with the practice and implication of their guidelines and frameworks and policies issues.
- the approach used by any of the selected external actors in mobilizing provisions and regulating sustainability reporting arena.
The SLR process we used is now provided in Figure 2. This helped us to identify the key articles that are relevant to our study. The selected articles are cited and their analysis were used in our study, particularly in section 4 to corroborate the understanding of the strategy and influence of the external actors (GRI, SASB and IIRC), which help to make sense of their behavior and how it is affecting the shape of the sustainability reporting regulatory arena. Further, additional information regarding how the documents and articles selected are used is now provided in the method section of the paper. Likewise, the list of the articles used is now provided as appendix for a sample (because of the length of the paper). Also, the eight black boxes that show the process of the systematic literature review has been adjusted and now readable (see Figure 2).
- The authors should explain the meaning of “ideology” (line 679) in the context of this work.
- The figures are sometimes not precise. According to the authors, figure 2 represents the NFR regulation market as a contested arena, but it is so different from figure 1 (source Renn, 1992) that it is difficult to interpret. Figure 3 should show the number of papers selected at each phase.
- subsection 6.1 (or only its title?) is missing
- In the section “Discussion and Conclusion”, I did not find the answer to the second research question: the authors seem more focused on the behavior of actors in the contested arena rather than on “the factors influencing the competition in the NFR regulation market”.
Authors’ Response: The meaning of “ideology” as used in the paper implies the specific interest each of the external actors is pursuing in the arena. We have redefined this in line with the research objective as the reviewer suggested. Also, all the figures that are not readable previously have been transferred into tables, and can now be read clearly. In addition, the arena concept has been re-mobilized, therefore, the position and definition of each actors in the arena has been re-defined in line with the explanation of arena concept by Rena (1992). This can be seen in section 2. Finally, the paper now focuses on the behavior and influence of the various external actors within the arena, and how this contributes to the future direction and harmonization of sustainability reporting regulation. This is in line with the reviewers’ comments for us to demonstrate and clearly show the practical implication of the paper and it connection with the theoretical framework.
Round 2
Reviewer 3 Report
The authors have done a good job. The manuscript has improved a lot in comparison with the first version.
I suggest only a careful re-reading due to some misprint (e.g. on page 29), long sentences (e.g. on page 1), and misuse of prepositions (e.g. on page 7) or, better, the authors could benefit from the help of a native English speaker.
Author Response
See file attached
